# Synthesis and Demulsification Properties of Poly (DMDAAC-*co*-DAMBAC) (9:1) Copolymer

**DOI:** 10.3390/polym15030562

**Published:** 2023-01-21

**Authors:** Xu Jia, Minghuan Qian, Wenhui Peng, Xiao Xu, Yuejun Zhang, Xiaolei Zhao

**Affiliations:** 1School of Chemistry and Chemical Engineering, Nanjing University of Science and Technology, Nanjing 210094, China; 2China National Quality Inspection and Testing Center for Industrial Explosive Materials, Nanjing University of Science and Technology, Nanjing 210094, China

**Keywords:** poly (DMDAAC-*co*-DAMBAC), copolymerization, weight-average molecular weight, demulsification performance

## Abstract

Utilizing the copolymerization modification of dimethyl diallyl ammonium chloride (DMDAAC), the high positive charge density of the copolymer could be maintained, thereby facilitating the deficiency of its monomer in the application. In this paper, poly (DMDAAC-*co*-DAMBAC) (9:1) was synthesized with an aqueous polymerization method using DMDAAC and methyl benzyl diallyl ammonium chloride (DAMBAC) as monomers and 2,2’-azobis [2-methylpropionamidine] dihydrochloride (V50) as an initiator. Targeted to the product’s weight-average relative molecular mass (*M_w_*), the response surface methodology (RSM) was used to optimize the preparation process. The optimal process conditions were obtained as follows: *w* (*M*) = 80.0%, *m* (V50):*m* (*M*) = 0.00700%, *m* (Na_4_EDTA):*m* (*M*) = 0.00350%, *T_1_* = 50.0 °C, *T_2_* = 60.0 °C, and *T_3_* = 72.5 °C. The intrinsic viscosity ([*η*]) of the product was 1.780 dL/g, and the corresponding double bond conversion (*Conv.*) was 90.25 %. Poly (DMDAAC-*co*-DAMBAC) (9:1) revealed a highest *M_w_* of 5.637 × 10^5^, together with the polydispersity index *d (M_w_/M_n_)* as 1.464. For the demulsification performance of simulated crude oil O/W emulsions, the demulsification rate of poly (DMDAAC-*co*-DAMBAC) (9:1) could reach 97.73%. Our study has illustrated that the copolymerization of DMDAAC and a small amount of DAMBAC with poor reactivity could significantly improve the relative molecular weight of the polymer, enhance its lipophilicity, and thus the application scope of the polymer.

## 1. Introduction

As a class of critical functional polymers, cationic quaternary ammonium salt polymers are extensively applied in industrial production and daily life, such as in petrochemical, wastewater treatment, textile printing and dyeing, and daily chemical industries [1,2,3,4]. As a typical cationic polyelectrolyte, dimethyl diallyl ammonium chloride (DMDAAC) is widely studied and applied [5,6,7,8]. Nevertheless, studies have illustrated that during the application process, the two methyl groups attached to the quaternary ammonium nitrogen of DMDAAC cause its polymer, poly (dimethyl diallyl ammonium chloride) (PDMDAAC), to have a weak affinity for fatty substances, which limits its application in the lipophilic field [9,10].

In the past few decades, research has been performed to modify DMDAAC’s structure and the preparation of its homopolymers and related copolymers to expand the application capability of this type of polyelectrolyte in the fields of sterilization, drug transportation, and fuel cells [11,12]. Lezov et al. [13] synthesized a random copolymer of dimethyl diallyl ammonium chloride (DMDAAC) and carboxyl betaine 2- (diallyl (methyl) ammonium) acetate (DAMA). Still, the optimization of the synthesis process and application efficiency of the copolymer have not been studied. Sun et al. [14] synthesized poly (methyl benzyl diallyl ammonium chloride) (PDAMBAC) and studied its staining properties. Zhang et al. [15] prepared poly (methyl propyl diallyl ammonium chloride) (PMPDAAC) and studied its demulsification properties. The above studies synthesized a series of allyl quaternary ammonium salts and their polymers with different substituents. At the same time, the substituents and, importantly, their impact on the polymerization rate and the characteristic viscosity of the products were discussed. These researchers realized that the lipophilicity of DAMBAC could be successfully improved by functional group modification. However, the researchers found that the homopolymers had a lower relative molecular mass. Based on previous research results, in some cases, the higher the comparable molecular group of polymers, the better the application effect, such as in the field of flocculation.

Due to the limited lipophilicity of diallyl quaternary ammonium salt homopolymers, to pursue better application efficiency, researchers usually use copolymerization with other lipophilic monomers to improve the application efficiency of this kind of polymer. Yang et al. [16] developed a new wastewater treatment agent (YL-7) using a cationic surfactant (LY) and flocculant (PDMDAAC/PAC) to treat aged oily wastewater (AOW) from the Tarim Oilfield in China. Wang et al. [17] synthesized a series of magnetic nanoparticles (MNPs) coated with PDMDAAC and fulvic acid (FA) (FeO/FA/PDMDAAC) by adjusting the concentration of dimethyl diallyl ammonium chloride (PDMDAAC) and applied them to the demulsification of cetyltrimethylammonium bromide (CTAB) stabilized micro lotion. The results showed that the oil-water separation efficiency (Es) increased with the positive charge on the surface of the demulsifier. This proved that the positive charge density plays a crucial role in the demulsification effect. Therefore, on the premise of maintaining a high cation in a polymer, the appropriate balance between the relative molecular mass of homopolymers and lipophilicity is worthy of consideration by researchers and is the basis of this study.

*N*,*N*-diallyl-*N*-methyl benzyl ammonium chloride (DAMBAC) was obtained based on DMDAAC by replacing a methyl group on quaternary ammonium nitrogen with benzyl, which can effectively improve the former’s deficiencies in lipophilicity. Jia et al. [8] determined that DMDAAC provides its homopolymer PDMDAAC, which had a good effect on the corresponding application sites. Still, in that study, the molecular weight of PDMDAAC was not high. At the same time, relevant research shows that improving the relative molecular weight could maintain specific application performance. Based on this, this paper was motivated to innovatively choose DMDAAC to copolymerize with DAMBAC to maintain the high cationic positive charge density and the stability of the five-membered ring quaternary ammonium salt in multiple conditions (pH and temperature). To prepare copolymers with higher relative molecular weights, the copolymerization ratio of poly (DMDAAC-*co*-DAMBAC) designed and synthesized in this paper was 9:1. The structures of DMDAAC and DAMBAC are shown in Figure 1.

Response Surface Methodology (RSM) is a statistical method that seeks the optimal solution by analyzing the fitted function relationship between a variable and a response value. This method has been widely used to optimize the polymerization reaction process for its efficient, intuitive, and superficial characteristics [18,19].

The purpose of this study was to synthesize poly (DMDAAC-*co*-DAMBAC) (9:1). The optimal process conditions were explored through Response Surface Methodology (RSM) and the establishment of the Box–Behnken mathematical relationship model [20,21]. The synthesized copolymer was characterized via FTIR, NMR, and a DAWN HELLOS Gel Permeation chromatography–multi-angle laser light scattering instrument (GPC-MALLS). According to the national standard, a laboratory-simulated O/W oil-bearing wastewater emulsion was used to compare the demulsification performance of poly (DMDAAC-*co*-DAMBAC) (9:1) with that of commercial demulsifiers and typical quaternary ammonium salts. Thus, the demulsification performance of poly (DMDAAC-*co*-DAMBAC) (9:1) was comprehensively evaluated.

Furthermore, the effect of relative molecular mass on the polymerization reactivity and demulsification performance was further studied and discussed. We suggest that the work described here could lay the foundation for developing diallyl cationic polymers and developing the structure and application properties of such copolymers.

## 2. Materials and Methods

### 2.1. Materials

V50 (mass fraction 99.0~100.0%, A.R.) (Nanjing Lanbai Chemical Co., Ltd., Nanjing, China), tetrasodium ethylenediaminetetraacetic acid (Na_4_EDTA) (mass fraction 99.0~100.0%, A.R.) (Sinopharm Shanghai Co., Ltd., Shanghai, China), sodium chloride (NaCl) (mass fraction ≥ 99.5%, A.R.) (Sinopharm Shanghai Co., Ltd., Shanghai, China), acetone (mass fraction ≥ 99.9%, A.R.) (Jiangsu Yonghua Fine Chemicals Co., Ltd., Nanjing, China), and nitrogen (99.9% by volume) (Nanjing No. 55 Institute, Nanjing, China) were obtained. DMDAAC solution (87.0~89.0% by mass) (Jiangsu Fumiao Technology Co., Ltd., Nanjing, China), DAMBAC solution (70.0~89.0% by group) (Nanjing University of Science and Technology, Nanjing, China), and distilled water (Nanjing University of Science and Technology, Nanjing, China) were also obtained.

### 2.2. Preparation of Poly (DMDAAC-co-DAMBAC) (9:1)

According to the molar ratio of DMDAAC and DAMBAC (*n*(DMDAAC):*n*(DAMBAC) = 9:1), three polymerization reactions were carried out. Two monomers with a total mass of 5.0 g were weighed into a four-necked flask. Certain amounts of Na_4_EDTA solution, distilled water, and initiator (V50) solution were added in sequence to adjust the monomer mass fraction (*w* (*M*)) and additional content (*m* (Na_4_EDTA):*m* (*M*)) and the initiator content (*m* (V50):*m* (*M*)) to the set value. Under nitrogen protection, the reaction system was stirred for 20–30 min to mix the materials evenly. Next, the reaction system was placed in a constant-temperature water bath. The temperature of the water bath was adjusted to change the polymerization initiation temperature (*T_1_*), the polymerization temperature (*T_2_*), and the ageing temperature (*T_3_*) so that the entire system was reacted at the three temperatures for three hours, respectively, which lasted for 9 h in total. After cooling, the material was discharged, and the characteristic viscosity ([*η*]) and monomer double bond conversion rate (*Conv.*) of the product were measured.

### 2.3. Single-Factor Exploratory Experiments

According to the preparation process in Section 2.2, the mass fraction of monomers, the mass fraction of auxiliaries and initiators in the monomers, and the temperature of the three stages of the reaction were changed. When *n*(DMDAAC):*n*(DAMBAC) = 9:1, the best result was obtained. The synthesis process conditions were: *w* (*M*) = 80.0%, *m* (V50): *m* (*M*) = 0.7 %, *m* (Na_4_EDTA):*m* (*M*) = 0.0035%, *T_1_* = 50.0 °C, *T_2_* = 60.0 °C, and *T_3_* = 72.5 °C. Meanwhile, the corresponding product had an intrinsic viscosity value of 1.78 dL/g and a double bond conversion (*Conv.*) of 90.25%.

### 2.4. Process Optimization Experiments

Based on the results of the single-factor exploration experiment in Section 2.3, the response surface method was used to optimize the process in this section. The Box–Behnken mathematical model in Design Expert 8.0 software was used to design a three-factor and three-level experiment with three critical factors, *w* (*M*) (*A_1_*), *m* (V50): *m* (*M*) (*B_1_*), and *m* (Na_4_EDTA): *m* (*M*) (*C_1_*), as response variables. The optimum feed ratio of the polymerization process was obtained by taking the product’s molecular weight per weight (*M_w_*) as the response value. Factor codes and levels of feeding ratio are shown in Table 1.

Based on the optimal feed ratio, a three-factor and three-level experiment was designed with the three-stage temperature *T_1_*(*A_2_*), *T_2_*(*B_2_*), and *T_3_*(*C_2_*) of the polymerization reaction as response variables, and a mathematical regression model was established. The optimal three-stage temperature data of the polymerization process were obtained by taking *M_w_* as the response value for optimization. The codes and levels of temperature factors are shown in Table 2.

### 2.5. Characterization

The instruments used in the characterization experiments included a Nicolet IS-10 Fourier infrared transform spectrometer (Thermo Fisher Scientific, Waltham, MA, USA), an Avance II 500 MHz Superconducting nuclear magnetic resonance spectrometer (Bruker Co., Ltd., Massachusetts, Switzerland), a DAWN HELLOS Gel Permeation chromatography-multi-angle laser light scattering instrument (GPC-MALLS) (Wyatt Technology Co., Ltd., Goleta, CA, USA), and a YYS-300E biological microscope (Shanghai Yiyuan Optical Instrument Co., Ltd., Shanghai, China). A capillary dilatometer (0.5–0.6 mm *Ubbelohde* viscometer) and mercury calibration volume (Shanghai Shenyi Glass Products Co., Ltd., Shanghai, China) were also used.

#### 2.5.1. Determination of Intrinsic Viscosity

According to “GB/T 33085-2016 Water Treatment Agent-Polydimethyldiallylammonium Chloride”, 0.1 g of poly (DMDAAC-*co*-DAMBAC) (9:1) was weighed into a 100 mL volumetric flask with an allowable error range of (±0.002 g). Then, 1.000 mol/L and 0.1000 mol/L NaCl solutions were selected to the volume and shaken until the copolymer was dissolved. The *Ubbelohde* viscometer was placed in a constant-temperature water bath at (30 ± 0.1) °C. At the same time, the retention time of the solution to be tested in the *Ubbelohde* viscometer was recorded and calculated using the single-point method.

#### 2.5.2. Determination of Double Bond Conversion

A total of 0.05 g of the product (the allowable range of weighing error was ±0.001 g) was considered and placed in a 500 mL conical flask. After that, 100 mL of distilled water was added to the conical flask and shaken well. This determination method was referred to as “GB/T 22312-2008 Plastics-Determination of Residual Acrylamide Content of Polyacrylamide”.

#### 2.5.3. Refinement of the Product

A certain amount of copolymer product was weighed into a 100 mL beaker, and a small amount of distilled water was added until the product was completely dissolved. Once the product was completely dissolved, acetone should have been added by degrees. As long as white viscous polymer was precipitated, acetone should have been added in 3–5 batches. Finally, the solution was stirred well until the polymer precipitated. After completing the above steps, we poured the acetone and repeated the above method three more times. After the above operations, the polymer was placed in a vacuum-drying oven at 40 °C. Without drying, the sample could not be taken out and pulverized to obtain the corresponding refined product. Usually, the purified product could be stored in a desiccator. After the product was refined, the solid content, intrinsic viscosity, relative molecular mass, and double bond conversion rate were used to determine the subsequent product structure and property characterization.

#### 2.5.4. Product Structure and Relative Molecular Mass Characterization

The structure and relative molecular mass of the refined product was characterized via infrared spectroscopy using the KBr method, in which the wave number ranged from 4000 cm^−1^ to 400 cm^−1^. The advanced product was tested using an NMR spectrum with D_2_O as solvent and tetramethylsilane as internal standard.

With *V* (methanol):*V* (water) = 75:25, 0.1 mol/L NaCl, and 0.01 mol/L NaH_2_PO_4_ as the mobile phase and polyethene glycol as the standard sample, GPC-MALLS was used to determine the relative molecular mass and distribution of the products under the conditions of 25 °C column temperature and 0.5 mol/min flow rate.

### 2.6. Demulsification Performance Experiment

The experiments were carried out in light of the Petroleum and Natural Gas Industry Standard of the People’s Republic of China (SY/T 5979-1993 Evaluation method of oil in water emulsions demulsified performance). Meanwhile, O/W-type oil-bearing wastewater emulsions were prepared, and a standard curve is presented in Figure 2.

According to Lambert Beer’s law, the absorbance of the tested substance at its maximum absorption wavelength is different at different concentrations. Therefore, to calculate the demulsification degrees, the absorbance of the emulsion before and after demulsification could be measured. Then, the oil content of the emulsion before and after demulsification could be obtained through the standard curve of oil. The specific process is shown in Equation (1).
*R* = (*w*_0_ − *w*_1_)/*w*_0_ × 100%(1)
where *R* is the demulsification rate, *w*_0_ is the oil content before demulsification, and *w*_1_ is the oil content after demulsification.

It could be seen from Figure 2 that the coefficient determination *R^2^* = 0.9993, indicating that the linear relationship between absorbance and oil content was excellent, which could be used for further experiments.

## 3. Results and Discussion

### 3.1. Results Analysis of Response Surface Methodology

#### 3.1.1. Surface Optimization of Feed Ratio Factors

The experimental design was optimized based on the feeding ratio factor, and the polymer product’s weight-average molecular weight was regressed using Design Expert 8.0 software. Afterwards, the fitting equation of the feeding ratio factor was obtained: *M_w_* = 63.11 + 1.07*A*_1_ + 1.96*B*_1_ + 0.059*C*_1_ − 1.46*A*_1_*B*_1_ + 0.49*A*_1_*C*_1_ + 1.49*B*_1_*C*_1_ − 9.8*A*_1_^2^ − 6.95*B*_1_^2^ − 8.13*C*_1_^2^. The overall analysis of variance for the factor model of the feed ratio is shown in Table 3.

Further analysis of Table 3 revealed that the model’s determination coefficient *R^2^* was 0.9132, indicating that the predicted value correlates well with the measured value. Nevertheless, this model could not represent 8.18 % of the total variation of the response value. The P-value indicated the model terms were significant. The “Lack of Fit F-value” of 1.24 implied a 40.43% chance that a “Lack of Fit F-value” this large could occur, which was insignificant. The above indicated that the predicted and measured values were well correlated, and the model was significant. The partial derivative of the fitting equation was equal to 0, and the best conditions for solving the equation were as follows: *A_1_* = 80.11%, *B_1_* = 0.71%, and *C_1_* = 0.004%.

We demonstrate the response surface diagram of the interaction of various factors on *M_w_* in the polymerization process in Figure 3. The analysis of Figure 3 could summarize the effect of the interaction between independent variables on the *M_w_* and obtain the best level of each feed ratio factor. It could be seen from Figure 3a that with the increase in *A_1_* and *B_1_*, the *M_w_* first increased and then decreased, growing relatively slowly. The *M_w_* reached the highest point when *A_1_* and *B_1_* were 80.11% and 0.71%. It is shown in Figure 3b that the *M_w_* first increased and then decreased with the increase in *A_1_* and *C_1_*. The *M_w_* reached the highest point when *A_1_* and *C_1_* were 80.11% and 0.004%. As seen from Figure 3c, with the continuous increase in *B_1_* and *C_1_*, the *M_w_* of the copolymer first increased and then decreased and finally reached the highest point when *B_1_* and *C_1_* were 0.71% and 0.004%, respectively.

Parallel verification experiments were performed on the optimal process conditions to verify the optimal conditions. The results are displayed in Table 4 below.

From the results in Table 4, it could be seen that through three parallel verification experiments, the repeatability of the synthesis process conditions was excellent, and the relative molecular mass of the obtained polymer ranged from 6.030 × 10^5^ g/mol to 6.450 × 10^5^ g/mol, with an average value of 6.240 × 10^5^ g/mol.

#### 3.1.2. Surface Optimization for Temperature Factors

The experimental design was optimized according to the temperature factor, and the Design Expert 8.0 software regressed the weight-average molecular weight of the polymer product. Afterwards, the fitting equation of the temperature factor was obtained as follows: *M_w_* = 71.63 + 4.57*A*_2_ − 8.84*B*_2_ + 0.08*C*_2_ + 1.18*A*_2_*B*_2_ − 0.21*A*_2_*C*_2_ + 2.21*B*_2_*C*_2_ − 12.60*A*_2_^2^ − 31.12*B*_2_^2^ − 14.25*C*_2_^2^. The overall variance analysis of the temperature factor model is shown in Table 5.

Further analysis of Table 5 revealed that the model’s determination coefficient *R^2^* was 0.9328, indicating that the predicted value correlated well with the measured value. Nevertheless, this model could not represent 6.72% of the total variation of the response value. The P-value indicated the model terms were significant. The “Lack of Fit F-value” of 5.42 implied a 6.80% chance that a “Lack of Fit F-value” this large could occur, which was insignificant. The above indicated that the predicted and measured values were well correlated, and the model was significant. The partial derivative of the fitting equation was equal to 0, and the best conditions for solving the equation were as follows: *A_2_* = 50.0 °C, *B_2_* = 60.0 °C, and *C_2_* = 72.5 °C.

Figure 4 shows the response surface graphs of the temperature factor of the weight-average molecular weight of poly (DMDAAC-*co*-DAMBAC) (9:1). The analysis of Figure 4 could summarize the interaction between independent variables on poly (DMDAAC-*co*-DAMBAC) (9:1). The influence of weight-average relative molecular mass obtained the best level of each temperature factor. It could be seen from Figure 4a that with the increase in *A_2_* and *B_2_*, the weight-average relative molecular weight of the copolymerized product first increased and then decreased. The molecular weight reached the highest point. It could be seen from Figure 4b that with the increase in *A_2_* and *C_2_*, the weight-average relative molecular weight of the copolymerized product increased and decreased.

Meanwhile, the increase and decrease trends were relatively gentle. When *A_2_* = 51.2 °C and *C_2_* = 72.4 °C, it reached the highest point. It could be seen from Figure 4c that with the continuous increase in *B_1_* and *C_1_*, the weight-average molecular weight of the copolymerization product first increased and then decreased. When *B_2_* = 59.2 °C and *C_2_* = 72.4 °C, it reached the highest point.

Parallel verification experiments were performed on the optimal process conditions to verify the correctness of the optimal conditions. The overall verification results are exhibited in Table 6 below.

The results in Table 6 revealed that three parallel verification experiments showed the perfect repeatability of the synthesis process conditions. Meanwhile, the relative molecular mass of the obtained polymer ranged from 6.260 × 10^5^ g/mol to 6.590 × 10^5^ g/mol, with an average value of 6.390 × 10^5^ g/mol.

According to the analysis of variance, under our experimental conditions, the order of the feed ratio factor influencing the *M_w_* of the copolymerization product was: *B_1_* > *A_1_* > *C_1_*. The initiator accounted for the monomer mass fraction (*B_1_*) on the copolymerization product weight average. The relative molecular mass had the most significant effect. Furthermore, the order of the polymerization process temperature influencing the *M_w_* of the copolymerized product was: *B_2_* > *A_2_* > *C_2_*. That is, the polymerization temperature (*B_2_*) significantly affected the *M_w_* of the copolymer.

The reason for this was that according to Arrhenius’s empirical formula and polymerization rate equation [20] when the initial monomer content is determined, the polymerization temperature and the effective concentration of the initiator in the system are essential factors affecting the polymerization rate. Nevertheless, the mass fraction of the monomer was also significant to the polymerization rate under certain conditions. As the polymerization temperature was higher than 60.0 °C, the average molecular weight of the product decreased gradually. The product’s average molecular weight also fell slowly when the monomer mass fraction was over 80.1%. In a specific range of conditions, when the polymerization temperature was about 60.0 °C, the monomer polymerization was promoted. If the temperature was above the optimum reaction temperature, the initiator and monomer reacted substantially to completion. Therefore, the intrinsic viscosity value of the product produced above this temperature did not change much.

The above analysis summarizes the optimization experiments on the polymerization process’s feed ratio and temperature factors. Overall, the results revealed that the best process conditions were: *w* (*M*) = 80.1%, *m* (V50):*m* (*M*) = 0.710%, *m* (Na_4_EDTA):*m* (*M*) = 0.00400%, *T_1_* = 50.0 °C, *T_2_* = 60.0 °C, and *T_3_* = 72.5 °C, corresponding theoretical weight-average relative molecular mass *M_w_* = 6.330 × 10^5^ g/mol.

### 3.2. Structural Analysis of Monomers and Copolymers

#### 3.2.1. FTIR Characterization

Two monomers and poly (DMDAAC-*co*-DAMBAC) (9:1) were subjected to FTIR measurement, and the obtained FTIR is exhibited in Figure 5. As seen in Figure 5, 3375 cm^−1^ was the expansion vibration absorption peak of O-H, and there was an absorption peak here to indicate the presence of water in the copolymer product. The peak around 2979 cm^−1^ was the C-H vibration absorption peak in the methyl group, and the peak at 1638 cm^−1^ was the vibration absorption peak of the C=C double bond. The height of the copolymer product at this place was significantly weakened, indicating that the double bond had been opened. The peak around 1467 cm^−1^ was the in-plane bending vibration absorption peak of methylene C-H connected to quaternary ammonium nitrogen. ( Figure 5b) A new absorption peak appeared at 1360 cm^−1^, which was a curved vibration absorption peak within the Methylene C-H surface of the polymer backbone, indicating that the monomer polymerized. The peak around 1223 cm^−1^ was the C-N telescopic vibration peak on quaternary ammonium nitrogen. Around 937 cm^−1^ was the off-plane bending vibration absorption peak of = C-H, and the height of the polymer at this place was significantly weakened. The mountains of about 700 cm^−1^ and 745 cm^−1^ were C-H surface bending vibrations on the benzene ring; the absorption peak of the monomer DAMAC was the strongest at this place, and the absorption peaks of the copolymer products at this place were enhanced, indicating that the benzyl content in the polymer became higher. The above results demonstrated the successful synthesis of the copolymer poly (DMDAAC-*co*-DAMBAC) (9:1).

#### 3.2.2. NMR Characterization

^1^H NMR spectra of DMDAAC (a), DAMBAC (b), and poly (DMDAAC-*co*-DAMBAC) (9:1) (c) are exhibited in Figure 6. An analysis of Figure 6 could be obtained: spectral (a), spectral (b), and spectral (c) were generated at about *δ* = 1.52 (1) and *δ* = 2.75 (2). *δ* = 1.52 (1) was the H atomic absorption peak of -CH_2_- on the polymer backbone, and *δ* = 2.75 (2) was the H atomic absorption peak on the five-membered ring -C=H-. The peaks of the spectral pattern (a) and spectral (b) *δ* = 6.00(1) and *δ* = 5.74(2) were H atomic absorption peaks on allyl double bonds, and the two absorption peaks associated with C=C bonds in spectrum (c) disappeared. The peak of the *δ* = 7.50(6) was the absorption peak of H atoms on the benzene ring, and the intensity of the absorption peak at the spectral Figure 6c increased. The above results revealed that the monomer had a polymerization reaction.

^13^C NMR spectra of DMDAAC (a), DAMBAC (b), and poly (DMDAAC-*co*-DAMBAC) (9:1) (c) are shown in Figure 7. An analysis of Figure 7 could be obtained: spectral graph (c) at *δ* = 27.32 (1), *δ* = 38.42 (2), and *δ* = 53.56 (3) around the emergence of new absorption peaks. *δ* = 27.32 (1) was the absorption peak of -CH_2_- on the polymer backbone, and the peaks at *δ* = 38.42 (2) and 53.56 (3) were the double absorption peaks of -C=H- and -CH_2_- on the pendant ring and the absorption peak of the C atom on the benzene ring at *δ* = 127.57~137.56 (6~8), respectively. The absorption peak intensity at the spectra in Figure 7c increased. Compared with spectra (a) and (b), the absorption peaks on the allyl C=C double bond in spectra (c) *δ* = 124.52 and 129.35 disappeared, and the above results showed that the monomers polymerized.

#### 3.2.3. GPC-MALLS Characterization

The characterization principle of GPC-MALLS is simple and easy to understand. The molecules are separated according to the relative molecular weight when the polymer solution flows through the chromatographic column (gel particles). The ones with higher molecular weights are in the front (that is, the flow time is short), and the ones with lower molecular weights are in the rear (that is, the flow time is long) [22,23].

As shown in Figure 8, according to the GPC-MALLS detection principle [24], the relative molecular weight of the refined sample poly (DMDAAC-*co*-DAMBAC) (9:1) ([*η*] = 2.26 dL/g, (*Conv.*) = 91.32%) was determined. Meanwhile, the flow time and cumulative distribution curve of the relative molecular weight of PMPDAAC were also obtained. According to the outflow time diagram, the outflow time of poly (DMDAAC-*co*-DAMBAC) (9:1) ranged from 12.4 min to 21.4 min, and the signal reached the highest value when the outflow time was 16.7 min. The relative molecular weight of poly (DMDAAC-*co*-DAMBAC) (9:1) polymer was 5.637 × 10^5^ g/mol, and the polydispersity index *d(M_w_/M_n_)* was 1.464. It could be seen that the relative molecular weight distribution of poly (DMDAAC-*co*-DAMBAC) (9:1) was relatively narrow, and the polymerization process conditions were appropriate.

According to the cumulative distribution curve of relative molecular weight, the homopolymers of PDMDAAC and PDAMBAC were dominant in poly (DMDAAC-*co*-DAMBAC) (9:1) polymers. At the same time, the copolymers between DMDAAC and DAMBAC contained less. The structure of poly (DMDAAC-*co*-DAMBAC) (9:1) is shown in Figure 9.

### 3.3. Effect of Copolymerization Ratio on Polymerization Activity

Different molar ratios of DAMBAC (substituent is benzyl) in the copolymer would lead to a tremendous difference in the characteristic viscosity of poly (DMDAAC-*co*-DAMBAC) copolymers. The optimum polymerization conditions of the copolymers with different copolymerization ratios and the homopolymerized products’ relative molecular weight and conversion were compared, as shown in Table 7.

It could be obtained from the analysis of Table 7 that in the copolymerization system, due to the difference in polymerization activity, the introduction of DAMBAC could complicate monomer polymerization. The system’s monomer polymerization activity would decrease, eventually, to the relative molecular weight of the copolymerization products. When the proportion of DAMBAC in the copolymer rose, the highest molecular weight of the copolymer product fell sequentially.

In the meantime, the copolymer’s highest intrinsic viscosity (relative molecular weight) was DMDAAC, the inherent viscosity was 3.44 dL/g, and the relative molecular weight was 1.144 × 10^6^ g/mol. As the proportion of DAMBAC in the copolymer increased, the intrinsic viscosity (relative molecular mass) of the copolymer product became lower and lower. It confirmed that as the proportion of DAMBAC in the copolymer increased, polymerization became more difficult. In addition, the enhancement of the steric hindrance effect made it challenging to prepare polymers with high molecular weights.

### 3.4. Effect of Intrinsic Viscosity (Relative Molecular Weight) on Demulsification Properties of Polymers

We also studied the optimum demulsification conditions and corresponding demulsification efficiency of the poly (DMDAAC-*co*-DAMBAC) (9:1) copolymer for O/W oil emulsion with different characteristic viscosities. The results are exhibited in Table 8.

It could be seen from Table 8 that under a specific copolymerization ratio, the level of intrinsic viscosity (relative molecular mass) had a significant influence on the demulsification process conditions of the copolymer. Overall, under the same copolymerization ratio, with the increase in the copolymer’s intrinsic viscosity (relative molecular weight), the dosage of the demulsifier and the demulsification temperature gradually decreased. Meanwhile, the demulsification time steadily shortened, and the demulsification rate also dropped. For the poly (DMDAAC-*co*-DAMBAC) (9:1) copolymer, when the intrinsic viscosity increased from [*η*] = 0.31 dL/g to [*η*] = 2.75 dL/g, the dosage decreased from 1.0 g/L to 0.1 g/L, the demulsification temperature decreased from 55.0 °C to 45.0 °C, and the demulsification time also reduced from 7 h to 5 h. The demulsification rate decreased slightly, from 97.73% to 96.81%. The main reason for this was that the molecular main chain length would rise with increased polymer characteristic viscosity (relative molecular weight). Meanwhile, the more polymer molecules that could achieve effective demulsification at the exact dosage, the higher the demulsification efficiency.

### 3.5. Comparison of Demulsification Performance with Other Demulsifiers

To further investigate the demulsification performance of the copolymer prepared in this paper, it was compared with the demulsification performance of two monomer homopolymers, PDMDAAC and PDAMBAC, and the commercially available demulsifier, BQ-05. The results are shown in Table 9. Influenced by the low intrinsic viscosity of PDAMBAC, homopolymers with intrinsic viscosity [*ƞ*] = 0.32 dL/g and [*ƞ*] = 0.33 dL/g were selected for comparison in this paper.

It could be seen from Table 9 that the demulsification rate of PDAMBAC was higher than that of the commercial demulsifier, BQ-05, and the demulsification rate of PDMDAAC was the lowest under the condition that the intrinsic viscosity value was close. The demulsification effect of the poly (DMDAAC-*co*-DAMBAC) (9:1) copolymer was not only higher than that of PDAMBAC, BQ-05, and PDMDAAC but also higher than the highest value of the BQ-05 demulsifier reported in the literature, which was 92.04% [25].

To compare the demulsification effects of different demulsifiers, PDMDAAC, BQ-05, and poly (DMDAAC-*co*-DAMBAC) (9:1) were selected for experiments, and the morphologies of oil droplets before and after the demulsification of the emulsion were photographed under a microscope. The results are shown in Figure 10.

It is generally believed that a physical or chemical reaction will occur due to the natural emulsifier effect between a demulsifier and an oil–water interface, which will adsorb on the oil–water interface and change the interface property, which will finally give rise to a smaller strength of the interface film and enable larger emulsion droplets to flocculate, merge, and finally break the emulsion [26]. It could be seen from Figure 9a that there were a lot of tiny oil droplets wrapped in water in O/W lotion before demulsification. From Figure 9c, the demulsification rate reached 65.12%. Since the demulsifier destroyed the interface facial mask of the O/W emulsion, the oil droplets in the water phase were significantly reduced, and the oil droplets were aggregated. From Figure 9d, the demulsification rate reached 93%. At this time, the water phase was photographed under a microscope, and it could be seen that the oil drops in the water phase had disappeared, achieving the effect of oil–water separation.

In conclusion, compared with DMDAAC, DAMBAC had more vigorous surface activity. Introducing a small amount of DMABAC (the substituent is benzyl) in a copolymer could effectively improve the lipophilicity of the polymer, reduce the emulsion’s interfacial energy [27], and improve the demulsification rate. The continuous introduction of the benzyl group made the steric hindrance play a significant role, which reduced the polymer’s effective interaction sites and contact area with the negatively charged suspending colloids in the oily wastewater, resulting in a slight decrease in the demulsification rate.

## 4. Conclusions

The preparation process was optimized by the response surface method, and the optimal process conditions were obtained as *w* (*M*) = 80.0%, *m* (V50):*m* (*M*) = 0.700%, *m* (Na_4_EDTA):*m* (*M*) = 0.00350%, *T_1_* = 50.0 °C, *T_2_* = 60.0 °C, and *T_3_* = 72.5 °C. The intrinsic viscosity of the product obtained under this process condition was 1.78 dL/g, and the double bond conversion (*Conv.*) was 90.25%. Through the characterization of the product structure and relative molecular mass, the results show that a poly (DMDAAC-*co*-DAMBAC) (9:1) product with a high, close molecular mass was synthesized, of which the relative molecular mass was *M_w_* = 5.637 × 10^5^ g/mol and the polydispersity index was *d(M_w_/M_n_)* = 1.464.

Since the substituent of DAMBAC was benzyl, as the proportion of DAMBAC in the copolymer increased, the steric hindrance effect significantly reduced the polymerization activity. The intrinsic viscosity (relative molecular mass) of the poly (DMDAAC-*co*-DAMBAC) (9:1) copolymer prepared in this paper was lower than that of PDMDAAC.

Compared with PDMDAAC and PDMABAC, the demulsification performance of poly (DMDAAC-*co*-DAMBAC) (9:1) copolymer was significantly improved, and its demulsification rate was up to 97.73% and 96.81%.

This work could lay a foundation for preparing diallyl cationic polymers and expanding the structure and application properties of such polymers.

## Figures and Tables

**Figure 1 polymers-15-00562-f001:**
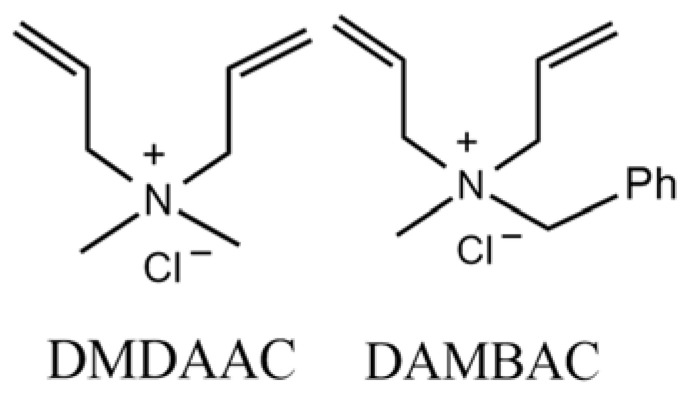
The structures of DMDAAC and DAMBAC.

**Figure 2 polymers-15-00562-f002:**
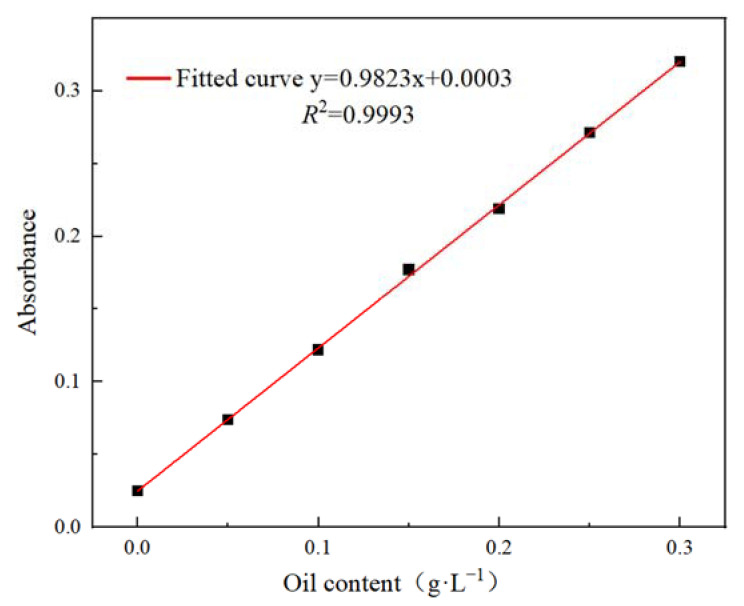
Standard curve of the oil.

**Figure 3 polymers-15-00562-f003:**
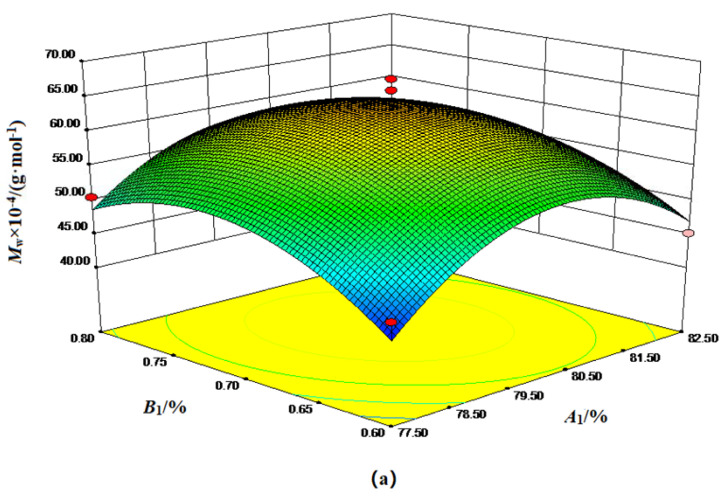
Response surface diagram of the interaction effect of various factors of the polymerization process feed ratio on *M_w_.* (**a**) Response surface diagram of the interaction between *A_1_* and *B_1_* of the polymerization process on *M_w_*; (**b**) Response surface diagram of the interaction between *A_1_* and *C_1_* of the polymerization process on *M_w_*; (**c**) Response surface diagram of the interaction between *B_1_* and *C_1_* of the polymerization process on *M_w_*.

**Figure 4 polymers-15-00562-f004:**
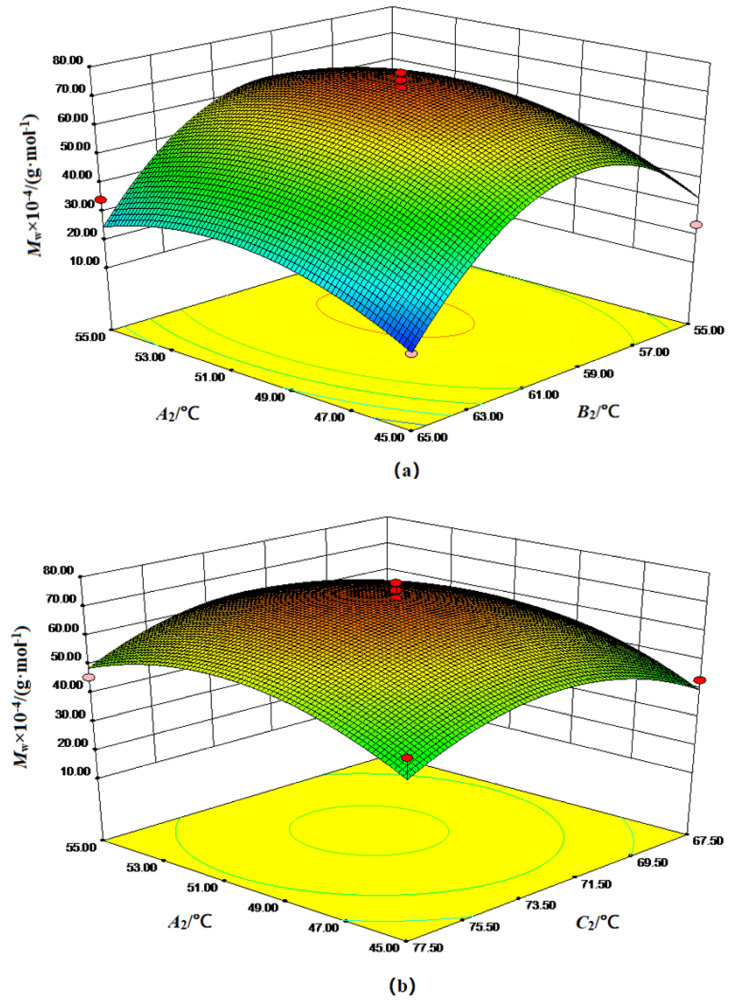
Response surface diagram of *M_w_* affected by various factors of polymerization process temperature. (**a**) Response surface diagram of the interaction between *A_2_* and *B_2_* of the polymerization process on *M_w_*; (**b**) Response surface diagram of the interaction between *A_2_* and *C_2_* of the polymerization process on *M_w_*; (**c**) Response surface diagram of the interaction between *B_2_* and *C_2_* of the polymerization process on *M_w_*.

**Figure 5 polymers-15-00562-f005:**
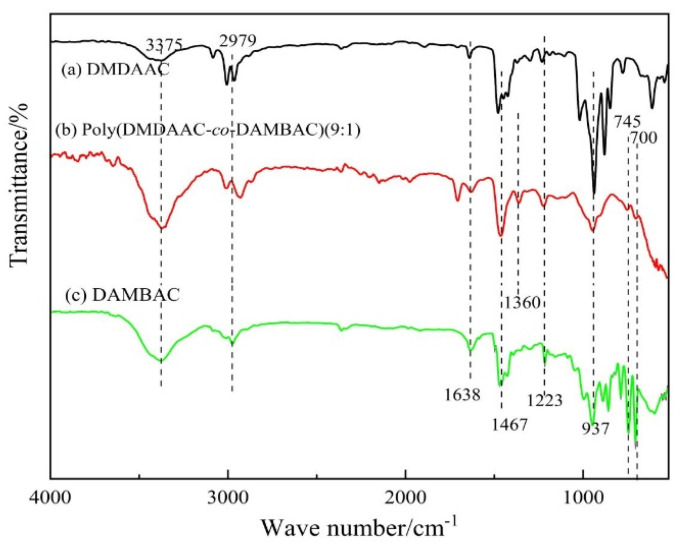
FTIR spectra of DMDAAC (**a**), DAMBAC (**b**), and poly (DMDAAC-*co*-DAMBAC) (9:1) (**c**).

**Figure 6 polymers-15-00562-f006:**
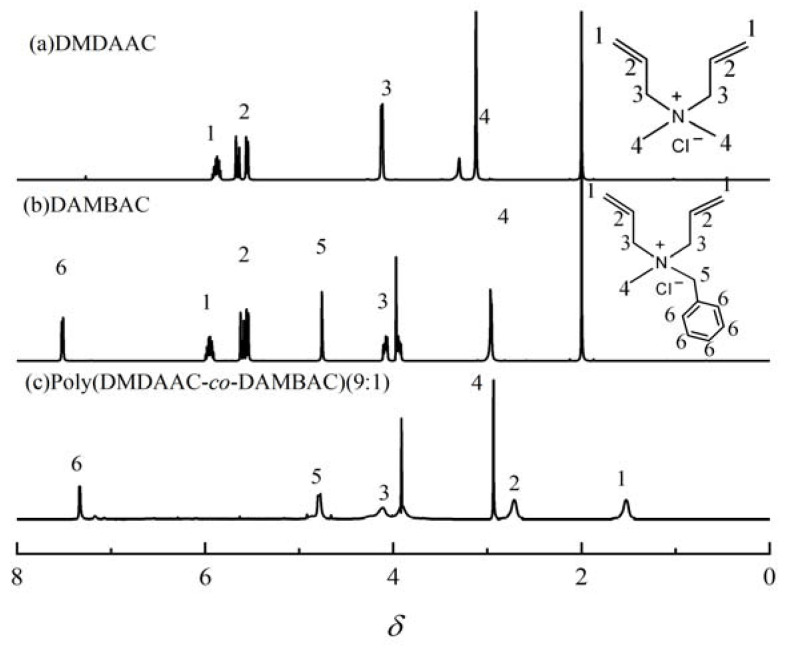
^1^H NMR spectra of DMDAAC (**a**), DAMBAC (**b**), and poly (DMDAAC-*co*-DAMBAC) (9:1) (**c**).

**Figure 7 polymers-15-00562-f007:**
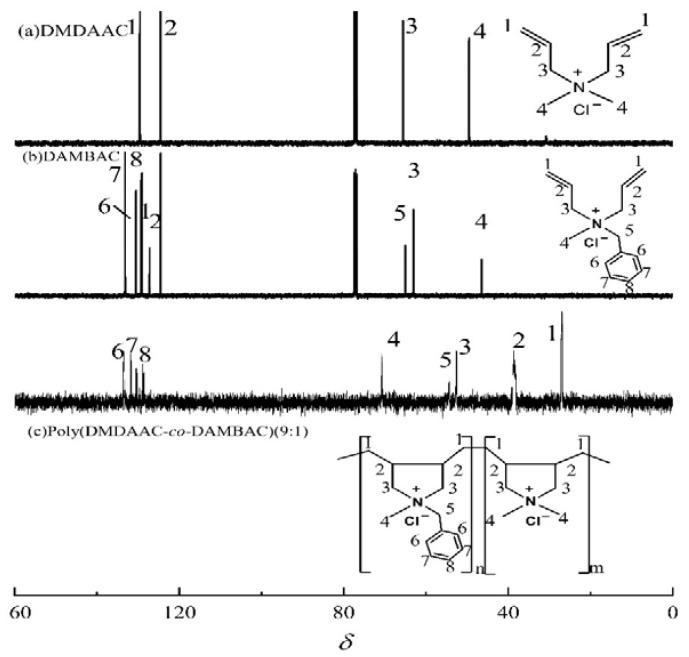
^13^C NMR spectra of DMDAAC (**a**), DAMBAC (**b**), and poly (DMDAAC-*co*-DAMBAC) (9:1) (**c**).

**Figure 8 polymers-15-00562-f008:**
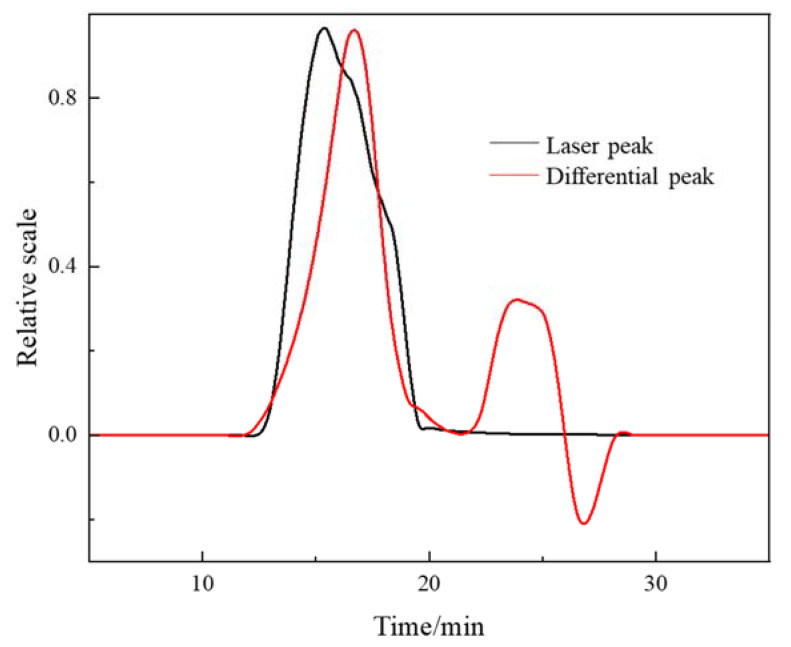
Cumulative distribution curves of poly (DMDAAC-*co*-DAMBAC) (9:1)’s efflux time and relative molecular mass.

**Figure 9 polymers-15-00562-f009:**
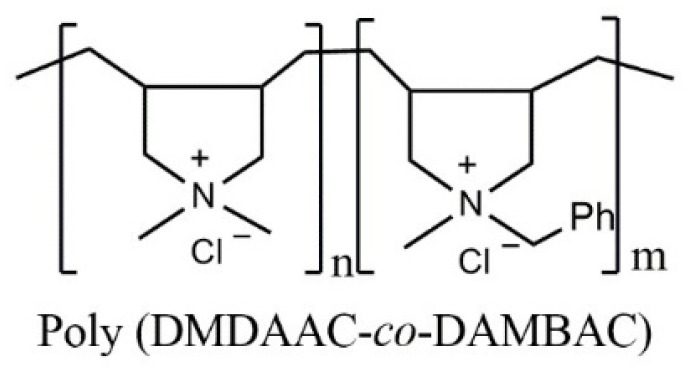
The structure of poly (DMDAAC-*co*-DAMBAC) (*n:m* = 9:1).

**Figure 10 polymers-15-00562-f010:**
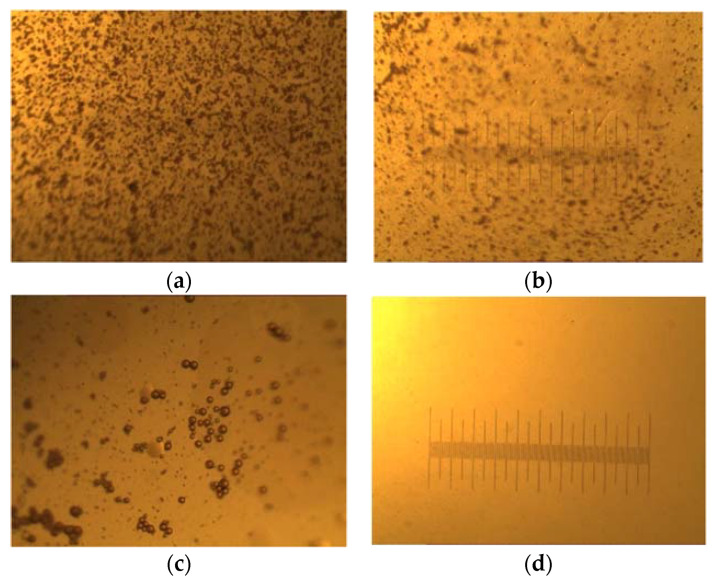
Micrograph of demulsification effects of different demulsifiers, of which the magnification was 40×. (**a**) Description of the sample before demulsification; (**b**) description of demulsification effect of PDMDAAC (0.32 dL/g); (**c**) description of demulsification effect of BQ-05; (**d**) description of demulsification effect of poly (DMDAAC-*co*-DAMBAC) (9:1).

**Table 1 polymers-15-00562-t001:** Feed ratio factor coding and level.

Factor	Code	Level		
		−1	0	1
*w* (*M*)/%	*A_1_*	77.50	80.00	82.50
*m* (V50)*:m* (*M*)/%	*B_2_*	0.60	0.70	0.80
*m* (Na_4_EDTA):*m* (*M*)/%	*C_3_*	0.002	0.004	0.006

**Table 2 polymers-15-00562-t002:** Temperature factor codes and levels.

Factor	Code	Level		
		−1	0	−1
*T*_1_/°C	*A* _2_	45.00	50.00	55.00
*T_2_*/°C	*B* _2_	55.00	60.00	65.00
*T*_3_/°C	*C* _2_	67.50	72.50	77.50

**Table 3 polymers-15-00562-t003:** Analysis of variance for factor model of feed ratio.

Source	SS	DF	MS	F	*p*-Value	Significance
Model	1045.42	9	116.16	8.18	0.0056	Significant
*A* _1_	9.10	1	9.10	0.64	0.4499	
*B* _1_	30.65	1	30.65	2.16	0.1852	
*C* _1_	0.03	1	0.03	1.94 × 10^−3^	0.9661	
*A* _1_ *B* _1_	8.56	1	8.56	0.60	0.4631	
*A* _1_ *C* _1_	0.96	1	0.96	0.07	0.8023	
*B* _1_ *C* _1_	8.85	1	8.85	0.62	0.4558	
*A* _1_ ^2^	404.26	1	404.26	28.47	0.0011	
*B* _1_ ^2^	203.14	1	203.14	14.30	0.0069	
*C* _1_ ^2^	278.54	1	278.54	19.61	0.0030	
Residual	99.41	7	14.20			
Lack of Fit	48.00	3	16.00	1.24	0.4043	Insignificant
Pure Error	51.41	4	12.85			
Total	1144.83	16				

**Table 4 polymers-15-00562-t004:** Parallel verification of optimal process conditions for optimization of feed ratio factors.

Number	*M_w_* × 10^−5^/(g·mol^−1^)	*Conv*./%	Remark
1	6.03	80.11	The first group
2	6.22	75.21	The second group
3	6.45	85.73	The third group
4	6.24	80.35	Average value

**Table 5 polymers-15-00562-t005:** Analysis of variance of the temperature factor model for the polymerization process.

Source	SS	DF	MS	F	*p*-Value	Significance
Model	6927.64	9	769.74	10.79	0.0024	Significant
*A* _2_	167.26	1	167.26	2.35	0.1695	
*B* _2_	625.52	1	625.52	8.77	0.0211	
*C* _2_	0.05	1	0.05	7.18 × 10^−4^	0.9794	
*A* _2_ *B* _2_	5.55	1	5.55	0.08	0.7884	
*A* _2_ *C* _2_	0.17	1	0.17	2.42 × 10^−3^	0.9622	
*B* _2_ *C* _2_	17.94	1	17.94	0.25	0.6314	
*A* _2_ ^2^	668.07	1	668.07	9.37	0.0183	
*B* _2_ ^2^	4078.03	1	4078.03	57.18	0.0001	
*C* _2_ ^2^	854.55	1	854.55	11.98	0.0105	
Residual	499.19	7	71.31			
Lack of Fit	400.69	3	133.56	5.42	0.0680	Insignificant
Pure Error	98.51	4	24.63			
Total	7426.83	16				

In Table 5, SS means sum of squares, DF means degrees of freedom, and MS means mean square. F value is the statistic of F test, that is, the ratio of the sum of squares of deviations between groups and within groups to the degree of freedom. Significance is the significance level corresponding to F statistic, and *p*-value means probability.

**Table 6 polymers-15-00562-t006:** Parallel verification of optimum process conditions for temperature factors.

Number	*M_w_* × 10^−5^/(g·mol^−1^)	*Conv*./%	Remark
1	6.32	86.22	The first group
2	6.59	88.65	The second group
3	6.26	87.29	The third group
4	6.39	85.05	Average value

**Table 7 polymers-15-00562-t007:** Summary of the best process conditions for copolymerization of monomers with different copolymerization ratios.

Samples	*w* (*M*) /%	*m* (V50)*:m* (*M*) /%	*m* (Na_4_EDTA)*:* *m* (*M*)/%	*T_1 _* /°C	*T_2 _* /°C	*T_3_* /°C	[*η*] /(dL·g^−1^)	*M_w_* × 10^5^ /(g·mol^−1^)	*Conv.* */%*
DMDAAC	65.00	0.70	0.0035	50.0	60.0	70.0	3.44	11.44	100.00
Poly (DMDAAC-*co*-DAMBAC) (9:1)	80.11	0.71	0.0040	51.2	59.2	72.4	2.27	6.59	88.65
DAMBAC	82.50	3.03	0.0147	61.8	70.6	85.0	0.38	0.79	75.12

**Table 8 polymers-15-00562-t008:** Summary of better demulsification process conditions of poly (DMDAAC-*co*-DAMBAC) (9:1) copolymers with different intrinsic viscosities.

Number	[*ƞ*]/(dL·g^−1^)	*D*/(g·L^−1^)	*T*/°C	*t*/h	*R*/%
1	0.31	1.0	55.0	7.0	97.73
2	0.98	0.5	50.0	5.0	97.20
3	2.75	0.1	45.0	5.0	96.81

**Table 9 polymers-15-00562-t009:** Comparison of demulsification performance with other demulsifiers.

Number	Demulsifier Sample	[*ƞ*]/(dL·g^−1^)	*D*/(g·L^−1^)	*t*/h	*T*/°C	*R*/%
1	BQ-05	-	5.0	4.0	60.0	84.12
2	PDMDAAC	0.32	2.0	4.0	55.0	37.34
3	PDAMBAC	0.33	2.0	5.0	55.0	91.56

## Data Availability

Data are contained within the article.

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
