# Peer review of "Synthesis and Demulsification Properties of Poly (DMDAAC-co-DAMBAC) (9:1) Copolymer"

_polymers, 2023, doi:10.3390/polym15030562_

Round 1
Reviewer 1 Report (Previous Reviewer 1)
The Authors updated the manuscript due to the previous version. Nevertheless, still, major revision is needed. Detailed comments were listed below:
- Figure 1 should be presented in the results and discussion section
- line 110 defines "V50"
- separate reagents from the apparatus and perform a detailed description of the setup of the experiment
- please see where some methods and types of equipment descriptions are duplicates - please change it (see the above comment)
- Please clarify how many samples you prepare
- Express what short names presented in table 3 means
Results and discussion
- please highlight which results are theoretical models and which are the experiment results
- line 382 - this part is difficult to read and I can not catch the meaning of the information presented here. Please change this part and make it clear
- line 481 please change the description marked in red - it is difficult to find what the Authors wanted to present here.
- due to the main topic of the article (suggested in the title) the demulsification section should be extended.
Author Response
Dear Editors and Reviewers:
Thank you for your letter and for the reviewers’ comments concerning our manuscript entitled “Synthesis and Demulsification Properties of Poly (DMDAAC-co-DAMBAC) (9:1) Copolymer” (ID: polymers-2081646). Those comments are all valuable and very helpful for revising and improving our paper, as well as the essential guiding significance to our research. We have studied the comments carefully and made corrections, which we hope meet with approval. Revised portions are marked in red in the manuscript. The significant modifications in the paper and the responses to the reviewers’ comments are as follows:
We tried our best to improve the manuscript and made some changes. These changes will not influence the content and framework of the paper. And here, we did not list the changes but marked them in red in the manuscript. We appreciate the Editors/Reviewers’ warm work earnestly and hope the correction will be approved. Once again, thank you very much for your comments and suggestions.
With our best regards
Xu Jia, Minghuan Qian, Wenhui Peng, Xiao Xu, Yuejun Zhang and Xiaolei Zhao
2022-12-08

Reviewer 2 Report (Previous Reviewer 2)
If editor and other reviewers agree that the paper is suitable for this journal I would recommend the paper for publication after the revision.
Author Response
Dear Editors and Reviewers:
Thank you for your letter and for the reviewers’ comments concerning our manuscript entitled “Synthesis and Demulsification Properties of Poly (DMDAAC-co-DAMBAC) (9:1) Copolymer” (ID: polymers-2081646). Those comments are all valuable and very helpful for revising and improving our paper, as well as the essential guiding significance to our research. We have studied the comments carefully and made corrections, which we hope meet with approval. Revised portions are marked in red in the manuscript. The significant modifications in the paper and the responses to the reviewers’ comments are as follows:
We tried our best to improve the manuscript and made some changes. These changes will not influence the content and framework of the paper. And here, we did not list the changes but marked them in red in the manuscript. We appreciate the Editors/Reviewers’ warm work earnestly and hope the correction will be approved. Once again, thank you very much for your comments and suggestions.
With our best regards
Xu Jia, Minghuan Qian, Wenhui Peng, Xiao Xu, Yuejun Zhang and Xiaolei Zhao
2022-12-08

Round 2
Reviewer 1 Report (Previous Reviewer 1)
The Authors improved the manuscript due to my comments.
Please consider for changing section 2.5 maybe characterization techniques or methods will be more suitable
This manuscript is a resubmission of an earlier submission. The following is a list of the peer review reports and author responses from that submission.
Round 1
Reviewer 1 Report
The Authors wanted to present the results of the evaluation of different factors for the selected properties of the studied copolymer.
Based on the presented results, it is difficult to find the main idea of the experiments, as well as a description of the performed research, that is not meaningful. Based on the presented description, it is impossible to verify the results of other researchers. Furthermore, only 17 references were cited, which suggests that the Author has not studied deeply the aim of the conducted research.
The manuscript should be rejected.
Detailed comments which might be helpful for the AUthors are below:
General remark: line numbering is helpful during the review
Abstract
- please try to make sentences shorter
- all short names should be explained ie. Mw
- what you mean by inspection index
- why did you choose presented parameters such as PDI, viscosity
- please describe in brief used research methods and present selected, the most important results
- briefly explain what you mean here by demulsification
- in the abstract, it should be highlighted that you study copolymers
Introduction
- please express what you mean by "...the most widely studied and applied"
- Response Surface Optimization (RSM) - RSO?
- a section which starts from "Poly (DMDAAC-co-DAMBAC) (9:1)..." - it looks like a discussion. Who performed this research - add references and insert to results and discussion. Furthermore, it also looks like the abstract. Please rewrite this section
Materials and methods
- separate materials from methods
- did you use FTIR with ATR?
- how many repetitions did you make for each composition
- characteristic viscosity ([η]) and monomer double bond conversion rate (Conv.) - how?
Please rewrite the materials and methods section:
- present methods in one section with subsections
- please provide detailed information about the samples' composition as well as their number
- what does mean Ukrainian viscometer? Give the full name of the equipment manufacturer, etc.
- please describe in detail how did you prepare the emulsion mentioned in section 2.6
- please describe in detail what kind of apparatus you used.
Results and discussion
- table 4 what you mean by "remark"
- how did you measure conversions
- please describe in detail where you present the prediction and where was experimental results
- figure 6 - enhance the quality of the figure - it is difficult to see the structures
- detailed description of GPC-MALLS is mandatory
- please add some comments about PDI
- how did you perform microscopy imaging?
- how did you evaluate demulsion properties - please add a detailed description
Conclusions
- there is no need to use numbers 1,2,3
References
11,12, 16, 17, - please try to translate it into English
Only 17 references are too low an amount
Author Response
Dear Editors and Reviewers:
Thank you for your letter and for the reviewers’ comments concerning our manuscript entitled “Synthesis and Demulsification Properties of Poly (DMDAAC-co-DAMBAC) (9:1) Copolymer” (ID: polymers-1982343). Those comments are all valuable and very helpful for revising and improving our paper, as well as the essential guiding significance to our research. We have studied the comments carefully and made corrections, which we hope meet with approval. Revised portions are marked in red in the manuscript. The significant modifications in the paper and the responses to the reviewers’ comments are as follows(Please see the attachment)
We tried our best to improve the manuscript and made some changes. These changes will not influence the content and framework of the paper. And here, we did not list the changes but marked them in red in the manuscript. We appreciate Editors/Reviewers’ warm work earnestly, and hope that the correction will be approved. Once again, thank you very much for your comments and suggestions.
With our best regards
Xu Jia, Minghuan Qian, Wenhui Peng, Xiao Xu, Yuejun Zhang and Xiaolei Zhao
2022-10-26
Abstract
- please try to make sentences shorter
Reply: Thank you for your suggestion. We have made the sentences more concise.
- all short names should be explained, ie. Mw
Reply: Thank you for your suggestion. We have explained all full names when they first appeared.
- what you mean by inspection index
Reply: Response surface methodology (RSM), also known as regression design, is a test design problem that needs to find the quantitative rule between the inspection index and each factor (rather than judging the element's significance and finding the best combination of each factor level). The inspection index is a quality index used to measure the testing effect.
- why did you choose presented parameters such as PDI, viscosity
Reply: Viscosity is an index parameter indicating the relative molecular weight under certain test conditions. PDI is a parameter representing the relative molecular weight. These two parameters are very important for polymers. They represent the molecular weight of products and also represent the concept of distribution.
- please describe in brief used research methods used and present selected, the most important results
Reply: Thank you for your suggestion. We have revised the statement and displayed the most important developments in the abstract.
- briefly explain what you mean here by demulsification
Reply: Demulsification refers to the process in which tiny droplets of the dispersed emulsion phase gather into clusters, form large droplets, and finally separate oil-water two-phase. In this paper, the lipophilicity of Poly (DMDAAC-co-DAMBAC) (9:1) was determined by demulsification performance.
The crude oil reserves in China are very rich. The exploitation and processing of crude oil are huge. In the process of crude oil exploitation, storage, transportation and processing, a large amount of wastewater containing crude oil will be generated, which is called crude oil wastewater. The crude oil wastewater has a high content of mineral oil, and is mainly composed of floating oil and emulsified oil, with high recycling value. Therefore, the demulsification treatment of crude oil wastewater is significant. It should not only improve the water quality, but also take into account the recovery of resources.
- in the abstract, it should be highlighted that you study copolymers
Reply: Thank you for your suggestion. In the abstract, we have been emphasizing copolymers.
Introduction
- please express what you mean by "...the most widely studied and applied"
Reply: Thank you for your requirement. We wanted to express here that this substance's research and application are extensive. Dimethyldiallylammonium chloride polymer is a water-soluble quaternary ammonium salt with linear structure. It has many advantages, such as high positive charge density, good water solubility, high efficiency, non-toxic, low cost, easy control of relative molecular weight, wide range of pH applications, stable cationic unit structure, etc. Therefore, it is an important cationic monomer.
- Response Surface Optimization (RSM) - RSO?
Reply: We are sorry for our improper use of words. RSM refers to Response Surface Methodology.
- a section which starts from "Poly (DMDAAC-co-DAMBAC) (9:1)..." - it looks like a discussion. Who performed this research - add references and insert to results and discussion. Furthermore, it also looks like the abstract. Please rewrite this section
Reply: This section have been rewritten.
Materials and methods
- separate materials from methods
Reply: We have made changes in Section 2.
- did you use FTIR with ATR?
Reply: No, We did not use FTIR with ATR. We used FTIR and NMR, which was enough for structural characterization
- how many repetitions did you make for each composition
Reply: Our experiment was a single point experiment without repetition. However, we repeated three times for the best point of process conditions.
- characteristic viscosity ([η]) and monomer double bond conversion rate (Conv.) - how?
Reply: Thank you for your question.
[η]: According to “GB/T 33085-2016 Water Treatment Agent-Polydimethyldiallylammonium Chloride”:
(1) Preparation of sample solution
Weigh 0.1-0.2 g of sample, record the mass m, add it into a 100mL volumetric flask, and then dilute it to the scale line with c (NaCl) = 1.0 mol/L solution at room temperature. After the sample is completely dissolved, shake the solution and filter it for standby.
Weigh 0.1-0.2 g of sample, record the mass m, add it into a 100-mL volumetric flask, and then dilute it to the scale line with c (NaCl) = 0.1 mol/L solutions at room temperature. After the sample is completely dissolved, shake the solution and filter it for standby.
(2) Determination steps
Shake up the filtrate, and use the filtrate to moisten and wash the Ubbelohde viscometer three times, especially the measuring tube, to ensure that the inner wall of the measuring tube is moistened and soaked by the filtrate many times. Then, pour the filtrate into the Ubbelohde viscometer and place the Ubbelohde viscometer vertically. The filtrate's liquid level is between the upper and lower limit labels of the primary liquid storage ball. At this time, the volume of the filtrate loaded is about 10ml. Place the Ubbelohde viscometer in a constant temperature (30 ± 0.1) ℃ water bath, preheat it for 10min, block the vent pipe, use an ear wash ball to suck the solution to the upper liquid storage ball of the measuring line, and the vent pipe is open to the atmosphere. When the fluid level in the measuring tube drops to the upper scale mark, start timing, and when it falls to the lower scale mark, stop timing. The time difference is t, repeat 3 to 4 times, and the error before and after shall not exceed 0.2 s. Finally, calculate the average value of t.
In addition, according to the above steps, determine the flow time of 1.0 mol/L and 0.1 mol/L NaCl aqueous solutions, which is t1.
Calculation formula of characteristic viscosity ([η]):
ηsp=t/(t1-1)
log[η]=log[ηsp/(m×G)]-k×ηsp
ηsp—Specific density.
[η] —Characteristic viscosity, dL/g.
t—The flow time of the sample solution, s.
t1—The flow time of 1.0 mol/L NaCl or 0.1 mol/L NaCl, s.
m—Sample mass, g.
G—Solid content of the sample (calculated by a mass fraction), %.
Where k is the coefficient, when ηsp< 0.3, k = 0.14. When 0.3<ηsp< 0.8, k = 0.12.
According to "GB/T 22312-2008 Plastics-Determination of Residual Acrylamide Content of Polyacrylamide":
Calculation formula of monomer double bond conversion rate (Conv.):
V1—Volume of Na2S2O3 standard solution consumed in blank test, mL.
V2—Volume of Na2S2O3 standard solution consumed by the sample, mL.
c—Concentration of Na2S2O3 standard solution, mol/L.
M1—The molar mass of the DMDAAC monomer is 161.67 g/mol.
M2—The molar mass of DAMBAC monomer is 237.77 g/mol.
Y—The molar ratio of DMDAAC in the copolymer.
M—Sample quality, g.
S—Solid content of sample, %.
Please rewrite the materials and methods section:
- present methods in one section with subsections
Reply: We have made some modification (Line 122 and 123).
- please provide detailed information about the samples' composition as well as their number
Reply: Thank you for your suggestion.
Table1 Surface optimization results for feed ratio factors in the polymerization process
|
Number |
A1/% |
B1/% |
C1/% |
Mw×10-5/(g·mol-1) |
Conv./% |
|
1 |
77.5 |
0.60 |
0.004 |
4.44 |
95.77 |
|
2 |
82.5 |
0.60 |
0.004 |
4.51 |
92.83 |
|
3 |
77.5 |
0.80 |
0.004 |
5.05 |
80.03 |
|
4 |
82.5 |
0.80 |
0.004 |
4.54 |
80.22 |
|
5 |
77.5 |
0.70 |
0.002 |
4.12 |
79.68 |
|
6 |
82.5 |
0.70 |
0.002 |
4.67 |
80.56 |
|
7 |
77.5 |
0.70 |
0.006 |
4.26 |
94.53 |
|
8 |
82.5 |
0.70 |
0.006 |
5.00 |
80.99 |
|
9 |
80.0 |
0.60 |
0.002 |
4.83 |
78.13 |
|
10 |
80.0 |
0.80 |
0.002 |
4.99 |
96.11 |
|
11 |
80.0 |
0.60 |
0.006 |
4.32 |
93.36 |
|
12 |
80.0 |
0.80 |
0.006 |
5.08 |
89.88 |
|
13 |
80.0 |
0.70 |
0.004 |
6.26 |
75.25 |
|
14 |
80.0 |
0.70 |
0.004 |
6.75 |
80.32 |
|
15 |
80.0 |
0.70 |
0.004 |
5.88 |
95.32 |
|
16 |
80.0 |
0.70 |
0.004 |
6.08 |
83.62 |
|
17 |
80.0 |
0.70 |
0.004 |
6.59 |
75.71 |
Table2 Surface optimization results for polymerization process temperature factors
|
Number |
A2/℃ |
B2/℃ |
C2/℃ |
Mw×10-5/(g·mol-1) |
Conv./% |
|
1 |
45.0 |
55.0 |
72.5 |
2.37 |
85.26 |
|
2 |
55.0 |
55.0 |
72.5 |
4.06 |
81.23 |
|
3 |
45.0 |
65.0 |
72.5 |
1.29 |
80.06 |
|
4 |
55.0 |
65.0 |
72.5 |
3.45 |
79.77 |
|
5 |
45.0 |
60.0 |
67.5 |
4.32 |
81.67 |
|
6 |
55.0 |
60.0 |
67.5 |
4.27 |
77.96 |
|
7 |
45.0 |
60.0 |
77.5 |
4.73 |
81.21 |
|
8 |
55.0 |
60.0 |
77.5 |
4.54 |
80.88 |
|
9 |
50.0 |
55.0 |
67.5 |
4.35 |
79.82 |
|
10 |
50.0 |
65.0 |
67.5 |
1.23 |
86.27 |
|
11 |
50.0 |
55.0 |
77.5 |
3.60 |
83.66 |
|
12 |
50.0 |
65.0 |
77.5 |
1.32 |
80.90 |
|
13 |
50.0 |
60.0 |
72.5 |
6.48 |
76.62 |
|
14 |
50.0 |
60.0 |
72.5 |
6.73 |
80.59 |
|
15 |
50.0 |
60.0 |
72.5 |
6.22 |
79.49 |
|
16 |
50.0 |
60.0 |
72.5 |
6.44 |
95.52 |
|
17 |
50.0 |
60.0 |
72.5 |
6.95 |
81.46 |
- what does mean Ukrainian viscometer? Give the full name of the equipment manufacturer, etc.
Reply: Thank you for your question. We are sorry for the wrong translation of “Ukrainian viscometer”. The viscometer used in the paper was “Ubbelohde viscometer.” We have added detailed information about Ubbelohde viscometer in Section 2.1.
- please describe in detail how did you prepare the emulsion mentioned in section 2.6
Reply: We have added the detailed information in section 2.6. (Line 199-213)
- please describe in detail what kind of apparatus you used.
Reply: We have added the detailed information in section 2.1. (Line 104-122)
Results and discussion
- table 4 what you mean by "remark"
Reply: We have conducted three parallel verification experiments and distinguished them by remarks.
- how did you measure conversions
Reply: We have put the answer in reply to this question “- characteristic viscosity ([η]) and monomer double bond conversion rate (Conv.) – how”
- please describe in detail where you present the prediction and where was experimental results
Reply: We did not involve prediction. Section 3 is all about the experimental results and the analysis of the results.
- detailed description of GPC-MALLS is mandatory
Reply: We have added a detailed description of GPC-MALLS. (Line 372-379)
- please add some comments about PDI
Reply: We have added comments about PDI. (Line 385-387)
- how did you perform microscopy imaging?
Reply: Thank you for your question. First, the four prepared samples were placed on the slide and covered the slide. Secondly, they were placed on the microscope stage and the magnification of the biological microscope was adjusted to 40 times. Finally, a record was taken after the imaging was clear.
- how did you evaluate demulsion properties - please add a detailed description
Reply: This paper compares the demulsification performance by calculating the demulsification rate and comparing the microscope pictures of different samples before and after demulsification. A detailed description has been added in section 2.4. (Line 199-213)
Conclusions
- there is no need to use numbers 1,2,3
Reply: We have deleted the numbers.(Line 483-498)
References
-11,12, 16, 17, - please try to translate it into English
Reply: We have translated them into English.
-Only 17 references are too low an amount
Reply: We have increased the number of references to 25.

Reviewer 2 Report
Poly (DMDAAC-co-DAMBAC) (9:1) is presented in the paper. It was synthesized by an aqueous polymerization method using DMDAAC and methyl benzyl diallyl ammonium chloride (DAMBAC) as monomers and 2,2'-azobis[2 methylpropionamidine] dihydrochloride (V50) as an initiator. Characterization of the polymer is presented. The paper could be considered for publication after revision. *Monomers DMDAAC and DAMBAC with ratio of 9:1 were used. Why this ratio was chosen ? *2,2'-azobis[2-methylpropionamidine] dihydrochloride (V50) was used as an initiator? Why this initiator was chosen as the most suitable for the polymerization? * Double bond conversion of monomers was 90.25 %. So, it seems that big amount of monomers is unreacted and the initiator is not suitable ? * The authors declare enhanced lipophilicity of new polymer and performance of the polymer. The advantages should be demonstrated in practical applications. * Also advantages and disadvantages of the new polymers should be compared with that of already published materials of this field. *Chemical structures of polymers should be demonstrated. *Practical applications of similar polymers should be also better described in introduction.Author Response
Dear Editors and Reviewers:
Thank you for your letter and for the reviewers’ comments concerning our manuscript entitled “Synthesis and Demulsification Properties of Poly (DMDAAC-co-DAMBAC) (9:1) Copolymer” (ID: polymers-1982343). Those comments are all valuable and very helpful for revising and improving our paper, as well as the essential guiding significance to our research. We have studied the comments carefully and made corrections, which we hope meet with approval. Revised portions are marked in red in the manuscript. The significant modifications in the paper and the responses to the reviewers’ comments are as follows(Please see the attachment)
We tried our best to improve the manuscript and made some changes. These changes will not influence the content and framework of the paper. And here, we did not list the changes but marked them in red in the manuscript. We appreciate Editors/Reviewers’ warm work earnestly, and hope that the correction will be approved. Once again, thank you very much for your comments and suggestions.
With our best regards
Xu Jia, Minghuan Qian, Wenhui Peng, Xiao Xu, Yuejun Zhang and Xiaolei Zhao
2022-10-26
- Monomers DMDAAC and DAMBAC with ratio of 9:1 were used. Why this ratio was chosen ? *
Reply: Thank you for your question. Our research group has made three kinds of copolymers in the early stage, namely Poly (DMDAAC-co-DAMBAC) (9:1), Poly (DMDAAC-co-DAMBAC) (5:5), and Poly (DMDAAC-co-DAMBAC) (1:9). According to their respective demulsification performance comparison, the copolymer with the best effect was described, and compared with the demulsifier on the market, highlighting its demulsification performance advantage under the copolymerization ratio (9:1).
- 2,2'-azobis[2-methylpropionamidine] dihydrochloride (V50) was used as an initiator? Why this initiator was chosen as the most suitable for the polymerization? *
Reply: Compared with peroxide initiators, azo initiators have the characteristics of high initiation efficiency, high molecular weight, and concentrated distribution. V50 is the most commonly used azo initiator for aqueous solution polymerization, and the half-life of the initiator is close to the polymerization temperature of the monomer.
- Double bond conversion of monomers was 90.25 %. So, it seems that big amount of monomers is unreacted and the initiator is not suitable ? *
Reply: Indeed, some monomers did not react. As for whether the initiator was inappropriate, it could only be said that it is challenging to coordinate the conversion of both monomers to 100 % in the case of copolymerization. In this paper, DMDAAC has been polymerized, but there was still a tiny amount of DAMBAC not polymerized. The polymerization activity of DMDAAC was higher than that of DAMBAC, and the reaction activation energy was also higher. Therefore, under the same polymerization process, the latter was difficult to fully react, and the conversion was slightly low. The initiators used in this paper have also been screened. Later, our research team will improve the degree of polymerization from the hybrid initiation system or by changing the polymerization method, such as the active monomer addition method.
- The authors declare enhanced lipophilicity of new polymer and performance of the polymer. The advantages should be demonstrated in practical applications. *
Reply: We have added some detailed descriptions. (Line 23-28, 449-481)
- Also advantages and disadvantages of the new polymers should be compared with that of already published materials of this field. *
Reply: The copolymer was designed to use the advantages of two monomers to synthesize the target copolymer. In this paper, DMDAAC monomer and modified aryl monomer were copolymerized to maintain 100% cationic degree so that each structural unit was a cationic degree, maintained relatively high molecular weight, and then increased lipophilicity through a certain proportion of lipophilic functional groups.
- Chemical structures of polymers should be demonstrated. *
Reply: We have shown the chemical structures of polymers in Figure 1.
- Practical applications of similar polymers should be also better described in introduction.
Reply: Thank you for your suggestion. We have added some detailed descriptions of practical applications of similar polymers in the introduction.

Round 2
Reviewer 1 Report
The Author partially improved the manuscript. Still, in my opinion, the manuscript quality is low and does not meet the Journals’ standards. Still, there is no clear data how many samples were studied?
Detailed comments are listed below:
The section which starts: "Poly (DMDAAC-co-DAMBAC) (9:1) was synthesized..." please do not describe your results in the introduction. This part looks like an abstract. Just general information is usually presented here.
Materials and methods
- methods need descriptions in detail i.e. conditions in which the samples were studied.
- once again question how did you perform FTIR experiments? Did your equipment has an ATR or you used a KBr matrix?
- figure 2 needs changes - please left only the trend line, not points matching
- The R2 parameter is not the same in the figure and the description. Just general information about Lamberts' law is not enough (and not necessarily in detail) to show how the demulsification index was calculated.
The section on surface optimization is difficult to understand? How many samples were used and analyzed here? What optimal factors are considered here? What about the prognosis and obtained results?
Please pay more attention to the description of what was presented in the figures and the text.
Furthermore, figs 2 and 9 need more improvements:
- fig2. the presented results are not clear. First, enlarge it, especially the fonts. then check the descriptions.
- fig. 9 there are no descriptions but emulsion structure in my opinion. Did you perform droplet size distribution or evaluate droplet size? What was the composition of the samples?
Reviewer 2 Report
If editor and other reviewers agree I could also recommend the paper for publication after the revision.